# Mitochondrial complex III subunit Qcr8 regulates the virulence and adhesion of *Candida albicans* by modulating mitochondrial function

Qianjun Zhao,[1,2,3] Xiaotian Huang,[2,4] Qiong Liu,[2,3] Yanli Cao,[2,3] Xiaomin Yu,[2,3] Yanling Liu,[2,3] Niya Hu,[2,3] Zhenning Han,[2,3] Junjun Tan,[2,3] Xuan Li,[2,3] Yemin Zhang,[2,3] Kai Wang,[2,3] Yuting Li,[2] Lingbing Zeng[1,2]

**ABSTRACT**  Invasive fungal infections caused by *Candida albicans* pose significant clinical challenges due to their high mortality rates and emerging drug resistance. In this study, we established Qcr8, an accessory subunit of mitochondrial complex III, as a critical virulence determinant in *C. albicans*. The deletion of *QCR8* markedly attenuated virulence in both *Galleria mellonella* and murine infection models, concomitant with impaired adhesion to biotic (human umbilical vein endothelial cells) and abiotic surfaces. Mechanistically, *QCR8* deletion disrupted mitochondrial homeostasis, evidenced by elevated reactive oxygen species levels, diminished membrane potential ($\Delta\Psi$m), and reduced ATP levels. Notably, cAMP levels decreased in mutant strains, resulting in the pronounced downregulation of Ras/cAMP/protein kinase A (PKA) pathway components (*RAS1*, *CYR1*, *TPK1/2*, *EFG1*, and *FLO8*), while exogenous cAMP supplementation partially restored the adhesion capacity. Our findings indicate that Qcr8 plays a vital role in mitochondrial complex III, highlighting its therapeutic potential as a fungus-specific drug target.

**IMPORTANCE**  We identified Qcr8, an accessory subunit of mitochondrial complex III for *C. albicans* full virulence. This study indicates that the accessory subunit of complex III, in addition to the structural subunits that have been previously the focus of study, also plays a significant role in the regulation of virulence. We also elucidated that Qcr8 promotes virulence via the Ras/cAMP/PKA pathway. Our findings established Qcr8 as a potential therapeutic target for treating *C. albicans* infections, which is particularly relevant, given the rising concern about antifungal resistance.

**KEYWORDS**  *Candida albicans*, mitochondria, virulence, adhesion, *qcr8*

In recent years, the global burden of invasive fungal infections (IFIs) has emerged due to the high prevalence of immunocompromised patients with various underlying diseases (1). *Candida albicans*, one of the most common invasive fungi (2), is a commensal yeast that colonizes the mucosal surfaces of the human oral cavity, gastrointestinal tract, genitalia, and skin (3). It can cause a wide range of diseases, from superficial mucosal infections to life-threatening systemic infections (4). Statistically, in different healthcare settings, the associated mortality rate of invasive candidiasis may range from 40% to 75%, with a total of approximately 250,000–700,000 cases of systemic infection and 50,000–100,000 deaths annually. Typically, *C. albicans* accounts for 40%–80% of *Candida* isolates recovered from hospitalized patients (5). Despite recent advancements in antifungal drug development, the mortality rate of IFIs remains high, reaching 40%–60% in intensive care unit patients with IFIs (6). This is primarily attributed to the complex pathogenic mechanisms of *C. albicans*, increasing drug resistance, and limitations of

Address correspondence to Lingbing Zeng, lingbing_zeng@163.com.

The authors declare no conflict of interest.

existing antifungal agents. Therefore, deciphering the pathogenic mechanisms of *C. albicans* and identifying novel therapeutic targets are crucial for improving the prognosis of IFI patients.

*C. albicans* demonstrates three critical steps in pathogenesis: adhesion, invasion, and host cell damage (7). Adhesion, the initial step of *C. albicans* infection, allows the fungus to adhere not only to cell surfaces but also to abiotic materials such as implanted medical devices (8). Previous studies have shown that the adhesion of *C. albicans* to vaginal epithelial cells is crucial for the pathogenesis of vulvovaginal candidiasis (VVC), and interfering with this adhesion can effectively alleviate VVC progression (9). Additionally, virulence factors associated with epithelial cell adhesion in *C. albicans* have been proposed as potential targets for antifungal drug development (10). Collectively, the adhesion ability of *C. albicans* is closely linked to its virulence, and adhesion-deficient strains often exhibit attenuated virulence.

As is well known, the Ras/cAMP/protein kinase A (PKA) signaling pathway plays a critical regulatory role in *C. albicans*, transmitting extracellular signals into the cell to modulate a series of virulence-related gene expressions and physiological processes. Pathway activation begins with the phosphorylation of Ras proteins, which, in turn, activate adenylate cyclase to promote cAMP synthesis (11). cAMP binds to the regulatory subunits of PKA, releasing and activating the catalytic subunits, which then phosphorylate downstream target proteins to regulate gene expression and cellular functions (12). Studies have shown that abnormalities in this pathway lead to reduced adhesion, invasion, hyphal formation, and attenuated virulence in *C. albicans* (13–15).

The ability of fungi to utilize various carbon sources plays a pivotal role in their colonization, influencing adaptability and pathogenicity. Fungi typically inhabit niches with limited glucose but abundant alternative carbon sources. When glucose is scarce, they utilize non-fermentable carbon sources, such as ethanol and acetate (16, 17). The mitochondrial respiratory chain is central to carbon source metabolism.

As the "energy factory" of the cells, mitochondria provide energy for various cellular activities. Changes in mitochondrial function often have fundamental and global impacts on cellular processes (18). The mitochondrial respiratory chain, which serves as the core site for energy conversion in mitochondria, consists of five multi-subunit complexes (complexes I–V). Among them, complex III (ubiquinol-cytochrome c reductase) plays a critical role in electron transport and proton transmembrane translocation, participating in oxidative phosphorylation to generate ATP (19). Complex III is composed of multiple subunits, including catalytic and accessory subunits, which act in concert to ensure its normal function. Previous studies have elucidated the functions of some subunits in mitochondrial respiratory chain complex III, mainly focusing on the catalytic subunits. For example, a lack of complex III in *Saccharomyces cerevisiae* leads to spermine toxicity (20). We previously studied Qcr7 in *C. albicans* and found that it was essential for its virulence and the stability of mitochondrial complex III (21). However, a similar accessory subunit in *C. albicans*, Qcr8, particularly its role in the pathogenic process, has not been examined. As an accessory subunit of mitochondrial respiratory chain complex III, Qcr8 may regulate the utilization of different carbon sources by *C. albicans*, thereby impacting its pathogenic process.

In this study, we focused on Qcr8 as an accessory subunit of mitochondrial respiratory chain complex III, investigating its effects on adhesion capacity, carbon source utilization, and the expression of genes associated with the Ras/cAMP/PKA signaling pathway. Through these investigations, we sought to uncover the critical role of *QCR8* in *C. albicans* pathogenesis, thus providing potential targets and theoretical foundations for the development of novel antifungal drugs.

## MATERIALS AND METHODS

### Structure prediction of multimeric proteins using AlphaFold3

AlphaFold3, a deep learning-based protein structure prediction tool, enables the structural modeling of multiple protein chains and the prediction of their interaction interfaces (22). First, the protein sequences of all subunits of the complex were obtained from the NCBI database (https://www.ncbi.nlm.nih.gov/) and used as input files for AlphaFold3 to construct the complex model to be predicted. For the generated models, the reliability of the predicted MEX3 and ELAVL1 complex was evaluated using the interfacial predicted TM (iPTM) score, expected position error matrix, and pLDDT. The iPTM score assesses the reliability of interactions in multimeric complexes, where a higher score indicates greater confidence in the interaction. Secondary structure data were then analyzed using the DSSP tool to investigate the impact of missing proteins on the structure. Finally, PyMOL was used to visualize the predicted complex interaction model obtained from AlphaFold3.

### Strain construction

First, knockout primers were designed for upstream amplification primers (up F and up R); downstream amplification primers (down F and down R); universal amplification primers for the nutritional selection marker genes *LEU2*, *HIS1*, and *ARG4* (primer2 and primer5); and verification primers (upcheck F, downcheck R, Leu left, Leu right, His left, His right, Target F, and Target R). Among them, the knockout primers up R and down F had homologous fragments of about 20 bp of the nutritional selection markers *LEU2*, *HIS1*, and *ARG4*, which were used for fusion PCR. Then, using the plasmids pSN40, pSN52, and pSN69 and the SN152 genome as templates, the *LEU2*, *HIS1*, and *ARG4* nutritional selection markers and the upstream and downstream of the target gene were amplified, respectively. *ARG4* was used as the nutritional marker for constructing the complementation strain. Next, fusion PCR was performed on the upstream and downstream with *LEU2* and *HIS1*, respectively. After successful fusion, the fusion fragment of the upstream and downstream of the target gene was transformed with *LEU2* into the SN152 strain (*leu2Δ/Δ*, *his1Δ/Δ*, and *arg4Δ/Δ*) using the standard *Candida* transformation method. After single clones grew, PCR verification was performed using the verification primers. After successful verification, the fusion fragment of the upstream and downstream was transformed with *HIS1* into a single target gene knockout strain, and screening for positive single-clone strains was carried out to complete the knockout of the target gene. The fusion fragment of the upstream and downstream, the target gene, and *ARG4* was transformed into the knockout strain, and screening for positive clone strains was carried out to complete the construction of the complementation strain. A schematic map is shown in the supplemental figure (Fig. S1).

### Animal experiment

Female BALB/c mice (4–6 weeks, 18–20 g) were purchased from Charles River and housed under specific pathogen-free conditions with a 12 h light/dark cycle. The day before the model was constructed, the target strains to be injected were inoculated into 5 mL of YPD medium (1% yeast extract, 2% peptone, and 2% glucose) and shaken overnight (14–16 h) (30°C, 200 rpm). The next day, the strain was transferred to 10 mL of YPD, and the strain concentration was adjusted to $OD_{600} = 0.1$ (30°C, 200 rpm, 5 h). After reaching the logarithmic phase ($OD_{600} = 0.6–0.8$), the strain was collected under 3,000 rpm for 5 min, washed twice with 1 mL of PBS, and resuspended with 1 mL of PBS. The mice were injected into the tail vein (100 µL of $5 \times 10^5$ colony-forming unit [CFU]/mL per mouse), and 100 µL of PBS was injected as a negative control group.

### Fungal burden

After 48 h of the animal experiment, target organs were aseptically harvested and homogenized in 1 mL of sterile PBS using a mechanical homogenizer under sterile

conditions until complete tissue dissociation. The homogenates were serially diluted 10-fold ($10^{-1}$ to $10^{-4}$) in sterile PBS. A fungal suspension (100 µL) from each dilution was plated onto YPD agar plates. After 48 h incubation at 30°C, viable colonies were counted and expressed as colony-forming units per gram of tissue.

## RT-qPCR

Total RNA was extracted from mouse tissues using an Animal Tissue/Cell Total RNA Extraction and Purification Kit (Servicebio, G3640-50T). The RNA concentration and purity were determined using a NanoDrop 2000 spectrophotometer (Thermo Fisher Scientific, USA), with A260/A280 ratios between 1.8 and 2.0. First-strand cDNA synthesis was performed with 500 ng of total RNA using a Prime Script RT Reagent Kit (Takara, RR092A) with oligo (dT) and random primers. Quantitative polymerase chain reactions were conducted in triplicate using TB Green Premix Ex Taq (Takara, RR420A) on a Quant Studio 5 Real-Time PCR System (Applied Biosystems, USA). Gene-specific primers were designed via Primer-BLAST on NCBI and validated for amplification efficiency. *GAPDH* and *18S* were used as endogenous controls. The thermal cycling protocol consisted of an initial denaturation at 95°C for 30 s, followed by 40 cycles at 95°C for 5 s and 60°C for 30 s. Primer specificity was confirmed using a melting curve analysis. Relative gene expression levels were calculated using the $2^{(-\Delta\Delta Ct)}$ method.

## Measurement of adhesive capacity under different carbon sources

A single colony of the target strain was inoculated into 5 mL of YPD and incubated at 30°C with 200 rpm shaking for 14–16 h. A 12-well plate was pre-coated with 5% (wt/vol) bovine serum albumin and placed in PBS solution at 4°C overnight with shaking. Cells were harvested via centrifugation at 3,000 rpm for 5 min and washed twice with 2 mL of PBS. The cell suspensions were adjusted to $OD_{600} = 0.5$ using modified spider medium (1% nutrient broth, 1% mannitol, and 0.002% dipotassium phosphate), where mannitol was replaced with specific carbon sources (glucose, sucrose, ethanol, and acetic). Then, 2 mL of a standardized fungi suspension was added to pre-coated 12-well plates, with three technical replicates per condition, and three wells containing sterile spider medium were supplemented with corresponding carbon sources as blank controls. The plates were subjected to adhesion at 37°C with 40 rpm orbital shaking for 90 min. Non-adherent cells were removed via washing with 1 mL of PBS three times. Adherent cells were fixed with 2 mL of methanol per well for 30 min and stained with 1% (wt/vol) crystal violet in 20% ethanol solution (2 mL/well) for 1 h. Excess stain was removed by lateral water rinsing until the eluent became colorless. The plates were inverted on absorbent paper for air-drying and photographed to document staining results. Bound dye was solubilized with 2 mL of glacial acetic acid per well under 30 min of shaking. Aliquots (200 µL) of the destaining solution were transferred to 96-well plates, and $OD_{620}$ values were measured using a microplate reader (Bio Tek Synergy H1, USA).

## HUVEC adhesion assay

HUVECs were activated and then adjusted to a concentration of $1 \times 10^6$ cells/mL using DMEM supplemented with 10% FBS. Then, 1 mL of cell suspension was seeded into each well of a 24-well plate, with six replicate wells per condition. The cells were cultured until reaching approximately 90% confluency. The strains were inoculated into 5 mL of YPD and incubated overnight at 30°C with shaking at 200 rpm. The overnight cultures were then diluted to an initial $OD_{600} = 0.1$ in YPD and grown to the logarithmic phase under the same conditions (4–6 h, 30°C, and 200 rpm). The cells were harvested, washed three times with PBS, and resuspended in serum-free DMEM to a final concentration of $1 \times 10^5$ CFU/mL. Then, 1 mL of the fungal suspension was added to HUVEC monolayers (pre-washed with PBS) for co-incubation. The coculture groups were divided by time points (1 and 2 h), with each group containing six wells: three control wells and three treatment wells. After incubation, the following steps were performed: 400 µL of 0.5%

Triton X-100 lysis buffer was added to lyse the cells for 10 min as a control group, and the supernatant was removed via centrifugation, followed by three washes with PBS as a treatment group. The cells were then lysed with 400 µL of 0.5% Triton X-100 for 10 min. The lysates from both groups were transferred to centrifuge tubes. After centrifugation, the pellets were resuspended in 1 mL of PBS, serially diluted, and plated onto YPD agar plates. The plates were incubated at 30°C for 48 h. Adherence rate (%) = (colony count of treatment group / colony count of control group) × 100%.

## Spot assay

A single colony of the target strain was inoculated into 5 mL of YPD and incubated overnight with shaking (30°C, 200 rpm). The strains were washed three times with PBS and resuspended in PBS to adjust the optical density to $OD_{600} = 1.0$. The suspension was then subjected to four successive 10-fold serial dilutions. Subsequently, 3 µL aliquots from each dilution gradient were spotted onto respective solid media plates. All plates were incubated at 30°C for 48 h in a constant-temperature incubator.

## Mitochondrial membrane potential assay in *C. albicans*

The mitochondrial membrane potential of *C. albicans* was assessed using a JC-1 Assay Kit (Beyotime Biotechnology, China). A single colony of the target strain was inoculated into 5 mL of YPD and incubated overnight with shaking (30°C, 200 rpm). The overnight culture was then diluted to an initial $OD_{600} = 0.1$ in 5 mL of YPD and grown with shaking (30°C, 200 rpm) for 4 h to reach the logarithmic phase. Concurrently, the JC-1 working solution was prepared by mixing 50 µL of JC-1 (200×) with 8 mL of ultrapure water under vigorous vertexing, followed by the addition of 2 mL of JC-1 staining buffer (5×). The working solution was stored at 4°C in the dark. Logarithmic-phase cells were harvested, washed three times with PBS, and adjusted to a concentration of $2 \times 10^6$ CFU/mL. Then, 500 µL of the cell suspension was combined with an equal volume of the JC-1 working solution, homogenized via gentle inversion, and incubated at 37°C for 20 min in the dark. For the positive control, wild-type cells were pre-treated with CCCP (a mitochondrial uncoupler) at a 1:1,000 dilution (final concentration: 10 µM) for 20 min, followed by PBS washing prior to staining. During the incubation period, JC-1 staining buffer (1×) was prepared by diluting 1 mL of 5× buffer with 4 mL of ultrapure water and stored on ice. Post-incubation, the stained cells were pelleted via centrifugation, washed twice with 1 mL of JC-1 buffer (1×), and resuspended in 1 mL of JC-1 buffer (1×). Fluorescence intensity was analyzed using a Cytomics FC500 flow cytometer, with signals captured in the PE (585/42 nm) and FITC (530/30 nm) channels.

## Measurement of intracellular reactive oxygen species levels

The intracellular reactive oxygen species (ROS) levels in *C. albicans* were quantified using a Reactive Oxygen Species Assay Kit (Beyotime Biotechnology). A single colony of the target strain was inoculated into 5 mL of YPD and cultured overnight with shaking (30°C, 200 rpm). The overnight culture was subcultured at an initial $OD_{600} = 0.1$ in 5 mL of fresh YPD and grown under identical conditions (30°C, 200 rpm) for 4 h to reach the logarithmic phase. Cells were harvested via centrifugation at 3,000 rpm for 5 min, washed three times with PBS, and resuspended in PBS to a final concentration of $5 \times 10^6$ CFU/mL. For staining, 1 µL of 20 mg/mL DCFH-DA solution was added to 1 mL of the cell suspension ($5 \times 10^6$ CFU/mL), followed by incubation at 37°C in the dark for 20 min (with three technical replicates per group). After incubation, the cells were pelleted via centrifugation, washed twice with PBS, and resuspended in 400 µL of PBS. Fluorescence intensity was analyzed using a Cytomics FC500 flow cytometer with the FITC channel (excitation/emission: 488/530 nm).

## Measurement of intracellular ATP and cAMP levels

The intracellular ATP and cAMP levels in *C. albicans* were measured as follows: a single colony of the target strain was inoculated into 5 mL of YPD and cultured overnight with shaking (30°C, 200 rpm). The overnight culture was subcultured at an initial $OD_{600}$ = 0.1 in 5 mL of YPD and grown under identical conditions (30°C, 200 rpm) for 4 h to reach the logarithmic phase. Cells were harvested via centrifugation at 3,000 rpm for 5 min, washed three times with PBS, and adjusted to a concentration of $1 \times 10^6$ CFU/mL in PBS. Subsequently, 1 mL of the cell suspension ($1 \times 10^6$ CFU/mL) was centrifuged at 3,000 rpm for 5 min, and the pellet was resuspended in 100 µL of PBS (with three technical replicates per group). The subsequent operations were conducted according to the instruction manuals of the ATP test kit (BacTiter-Glo Microbial Cell Viability Assay; Promega, TB337, USA) and the cAMP test kit (cAMP-Glo Assay; Promega, V1501).

## Statistical analysis

All experiments were performed in triplicate. Data are presented as mean ± standard deviation from the means. Analyses were performed using GraphPad Prism 10.1.2 software. An analysis of survival was completed using the log-rank chi-square test. Significant differences were calculated using Student's *t*-tests when comparing two groups and a two-way ANOVA when comparing three or more groups.

## RESULTS

### Qcr8 and Qcr7 are two distinct functional accessory subunits of complex III in *C. albicans*

Respiratory complex III in the model fungus *Saccharomyces cerevisiae* comprises 10 subunits, namely, 3 catalytic subunits (Cytb, Cyt1, and Rip1), 5 accessory subunits (Qcr6, Qcr7, Qcr8, Qcr9, and Qcr10), and 2 core assembly subunits (Cor1 and Cor2) (23, 24). In our previous study, we explored the function of Qcr7, one of the accessory subunits of complex III in *C. albicans*. However, the adjacent subunit, Qcr8, was not examined. Indeed, we identified *C. albicans* Orf19.4490.2 as *C. albicans QCR8* through protein sequence alignment; it shares 62.11% identity with *S. cerevisiae QCR8*, as well as sharing 39.25%, 30.84%, and 15.89% identity with *Neurospora crassa*, *Aspergillus fumigatus*, and *Homo sapiens QCR8* (Fig. 1A). In addition, a comparative sequence analysis between *C. albicans QCR8* and our previously characterized *QCR7* demonstrated striking divergence, with merely 11.81% sequence identity (Fig. 1B), suggesting non-overlapping functional roles for these two subunits. To test this hypothesis, we performed structural modeling with AlphaFold3 to analyze the key states of the full complex III structure (Fig. 1C) and the complex missing Qcr7 (Fig. 1D), missing Qcr7 but with the addition of Qcr8 (simulated through insertion of an additional *QCR8* sequence, Fig. 1E), and missing Qcr8 (Fig. 1F). The four groups of model prediction results showed that the iPTM scores ranged from 0.83 to 0.85. (The iPTM is used to assess the accuracy of multi-polymer interface prediction, with the range being from 0 to 1. The closer it is to 1, the more accurate the interface prediction is. It is mainly used for predicting the structure of complexes.) The pTM scores ranged from 0.83 to 0.85 (The pTM reflects the global prediction confidence of the overall structure, with a range of 0–1. The closer it is to 1, the more reliable the predicted structure is. Generally, values greater than 0.8 indicate a high level of confidence in the prediction.), and the pLDDT values were between 84.1 and 84.4 (The pLDDT reflects the prediction accuracy of each amino acid, and it ranges from 0 to 100. A value below 50 indicates a very inaccurate prediction or an intrinsically disordered protein; a value below 70 indicates low confidence; and a value above 90 indicates extremely high confidence.) (Fig. S3). This indicates that the folded structures of the four groups of prediction models were highly accurate; the relative positions of subunits were also reliably predicted; and the overall structures had high confidence. In the structure prediction alignment error (PAE) of the four groups of prediction models, the interaction interfaces in the self-related regions of the complex (dark green/green parts on the

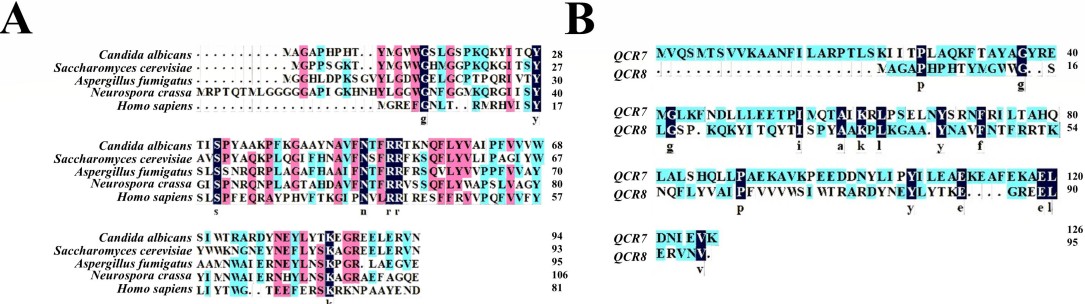

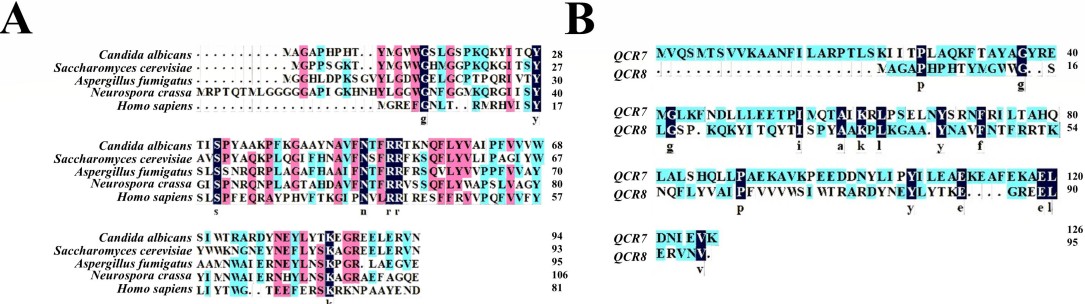

**FIG 1** Qcr8 and Qcr7 are two distinct functional accessory subunits of complex III in *C. albicans*. (A) Protein sequence alignment of Qcr8 from *Candida albicans*, *Saccharomyces cerevisiae*, *Neurospora crassa*, *Aspergillus fumigatus*, and *Homo sapiens*. (B) Protein sequence alignment between Qcr7 and Qcr8 in *C. albicans*. (C) Predicted complete structure of complex III in *C. albicans*. (D) Predicted structural changes in complex III in the *QCR7* deletion mutant. (E) Predicted structural changes in complex III in *QCR8* overexpression in the *QCR7* deletion mutant. (F) Predicted structural changes in complex III in the *QCR8* deletion mutant.

diagonal) had low PAE values, suggesting that the models showed stable predictions for the internal structures of the interaction interfaces of each chain (Fig. S4). The relative positions of the interaction regions of the complex as a whole were relatively balanced, and the relative positions of the interactions between complexes were stable, which also indicates high confidence.

When analyzing the impact of missing Qcr7 and Qcr8 on the structure, DSSP was used to analyze secondary structure changes. The missing Qcr8 in the secondary structure diagram led to the loss of the Qcr7 α-helix region, while the missing Qcr7 had little effect on Qcr8, suggesting that the missing Qcr8 may alter the stability, activity, or interactions with other subunits of the complex (Fig. S5). The full complex had 31 hydrogen bonds, and, when Qcr7 was missing, this number slightly decreased to 30 with a similar pattern, indicating minimal structural perturbation. However, the missing Qcr8 caused a sharp drop in the number of hydrogen bonds to 11, showing severe structural damage. Adding Qcr8 to the complex missing Qcr7 increased the number of hydrogen bonds to 34, but the pattern resembled that of the missing Qcr7 state rather than that of the full complex. In terms of the number of salt bridges, the complete complex had 4, involving Qcr8 with Qcr6 and Cor1. When Qcr7 or Qcr8 was missing, this number decreased to 3, but the latter's interaction partners became Qcr7 with Cytb and Cyt1. Adding QCR8 still left three (salt bridges) without restoring the original pattern. π-Stacking interactions remained consistent in the full and Qcr7 missing complexes (involving Cyt1 and Qcr8); however, when Qcr8 was missing, they shifted to Cytb and Qcr7, and adding Qcr8 did not change this trend. The number of π-cation interactions decreased from two in the full complex to one in the Qcr7 missing complex and none in the Qcr8 deficiency. Adding Qcr8 restored the number to 2, but the specific pattern did not fully match that of the complete complex (Table S1).

Functionally, missing Qcr7 had little impact on the integrity and function of the complex, while missing Qcr8 severely disrupted the interaction network, potentially causing significant functional impairment. Although adding Qcr8 provided some compensation, it could not fully restore the original structure and function, indicating that Qcr8 cannot completely substitute for QCR7. The above results suggest that *C. albicans* Qcr7 and Qcr8 may play distinct roles, and we next conducted in-depth research on QCR8.

## QCR8 affects the virulence of *C. albicans*

In this study, we constructed a *QCR8* mutant (*qcr8Δ/Δ*) and complemented strains (*qcr8Δ/QCR8*); then, we employed two distinct *C. albicans* infection models to assess the virulence of the wild-type strain (SN250), *qcr8Δ/Δ*, and *qcr8Δ/QCR8*. In the *Galleria mellonella* infection model, all *Galleria mellonella* individuals infected with the wild-type strain died within 12 h, while those infected with *qcr8Δ/QCR8* exhibited a survival rate of less than 25% at 12 h and complete mortality by 24 h. In contrast, both the PBS control and *qcr8Δ/Δ* mutant groups showed no mortality during the 48 h observation period, maintaining a 100% survival rate (Fig. 2A). Similar results were observed in the murine systemic *C. albicans* model. A survival curve analysis revealed no mortality in the PBS control or *qcr8Δ/Δ* mutant group over a 21-day observation period, with survival rates remaining at 100%. Conversely, all mice infected with the wild-type or *qcr8Δ/QCR8* strains succumbed to infection within 11 days (Fig. 2B). A histopathological examination of kidney tissues via hematoxylin–eosin staining demonstrated extensive inflammatory cell infiltration in the mice infected with the wild-type or *qcr8Δ/QCR8* strains. In contrast, no significant inflammatory infiltration was observed in the PBS control or *QCR8* mutant group (Fig. 2C). Furthermore, the fungal burden in the kidneys infected with the *qcr8Δ/Δ* mutant was significantly reduced compared to those infected with the wild-type or complemented strains (Fig. 2D). Additionally, an RT-qPCR analysis of kidney tissues revealed markedly elevated expression levels of the inflammatory cytokines *MACTIN*, *KIM1*, *TNF-α*, and *IL-1β* in the mice infected with the wild-type or *qcr8Δ/QCR8* strains, while minimal upregulation was detected in the PBS control and *qcr8Δ/Δ* groups (Fig.

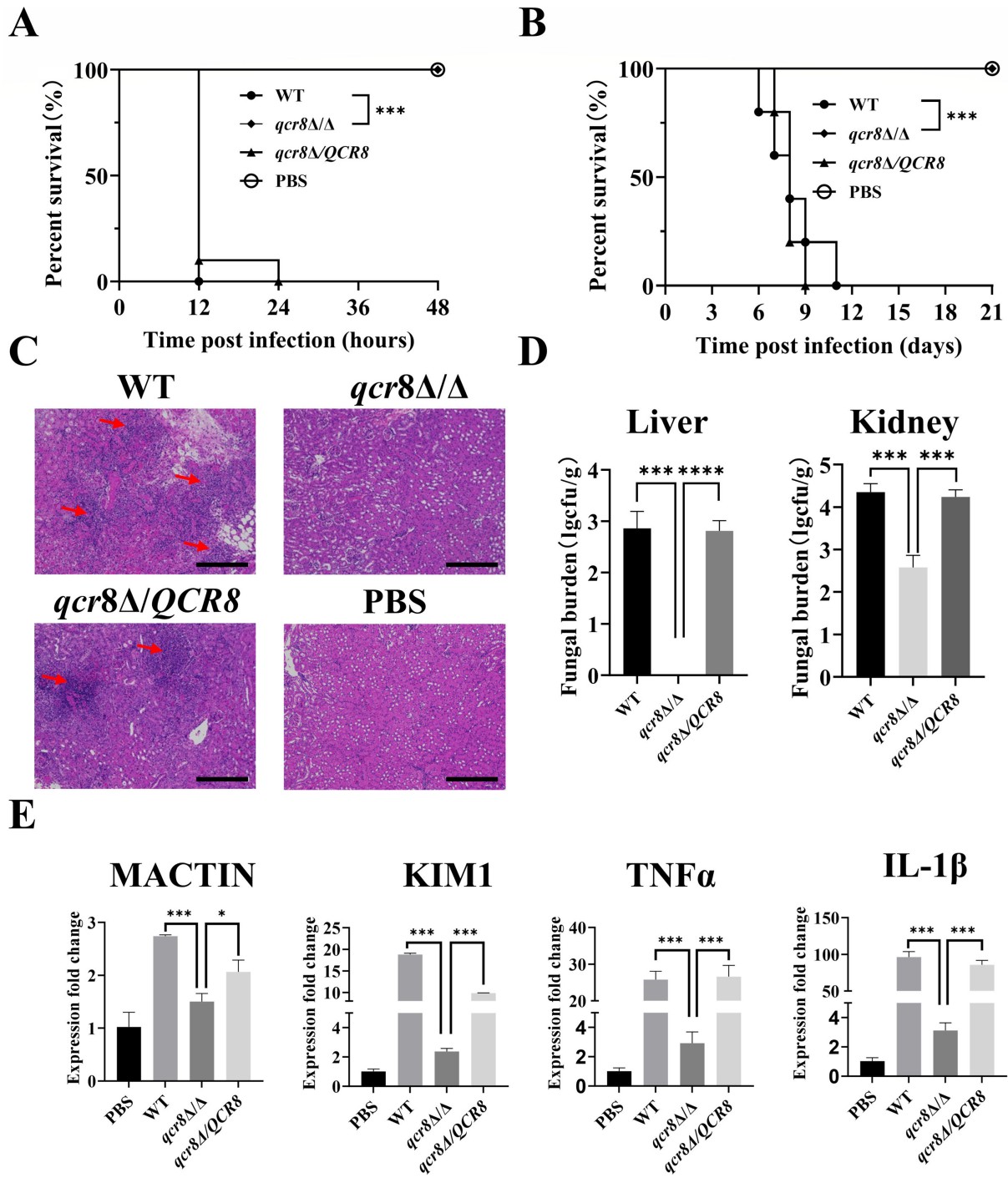

**FIG 2** *QCR8* affects the virulence of *C. albicans*. Wild-type *C. albicans* (WT), *qcr8Δ/Δ* (*QCR8* knockout), *qcr8Δ/QCR8* (*QCR8* complemented), PBS (control group). (A) Survival curves of Galleria mellonella infected with *C. albicans* strains (*n* = 10). ****P* < 0.001. (B) Survival rates of BALB/c mice monitored for 21 days post-infection (*n* = 5). ****P* < 0.001. (C) Representative hematoxylin–eosin-stained kidney sections at 48 h post-infection. Arrows indicate inflammatory cell infiltration. Scale bar = 300 µm. (D) Fungal burdens in the liver and kidney tissues of mice infected with the WT, *qcr8Δ/Δ*, and *qcr8Δ/QCR8* strains (*n* = 3). ****P* < 0.001, *****P* < 0.0001. (E) Relative fold change of inflammatory markers (*MACTIN*, *KIM-1*, *TNF-α*, and *IL-1β*) in kidney tissues measured using RT-qPCR (*n* = 3). **P* < 0.05, ****P* < 0.001. Data are presented as mean ± SD.

2E). Collectively, these findings demonstrate that *QCR8* deletion leads to significant virulence attenuation in both *Galleria mellonella* and murine infection models, indicating that *QCR8* plays a critical role in mediating the pathogenicity of *C. albicans*.

## QCR8 affects the adhesion capacity of *C. albicans*

Adhesion to biotic and abiotic surfaces is a critical virulence trait of fungal pathogens. In this study, we employed 12-well polystyrene plates to simulate the adhesion capacity of *C. albicans* to abiotic surfaces. Under spider medium at 37°C as induced conditions, crystal violet assays revealed that the adhesion capacity of the *qcr8Δ/Δ* mutant was significantly reduced compared to that of the wild-type *C. albicans* (WT) and *qcr8Δ/QCR8* complemented strains (Fig. 3A and B). The results revealed that the adhesion ability of *qcr8Δ/Δ* decreased compared to that of both WT and *qcr8Δ/QCR8* (Fig. 3C). These findings suggest that *QCR8* plays a crucial role in the adhesion process of *C. albicans*. Next, we wanted to investigate the factors causing the decrease in the adhesion ability of the *QCR8* mutant.

## QCR8 affects carbon source utilization in *C. albicans*

Glucose, the primary carbon source utilized by *C. albicans*, is typically present at low concentrations in host tissues. Under such conditions, *C. albicans* adapts by using alternative carbon sources for energy production. As an auxiliary subunit of complex III in the respiratory chain of *C. albicans*, *QCR8* may play a critical role in carbon source utilization. To investigate whether *QCR8* influences the utilization of different carbon sources, we supplemented carbon-free YEP medium with 2% glucose, sucrose, ethanol, or acetate. Glucose and sucrose served as fermentable carbon sources, while ethanol and acetate served as non-fermentable carbon sources. Spot assays demonstrated that the growth of the *qcr8Δ/Δ* mutant was slightly impaired on solid plates containing fermentable carbon sources (glucose and sucrose), whereas it was severely compromised on those containing non-fermentable carbon sources (ethanol and acetate) (Fig. 4A). These results indicate that *QCR8* deletion significantly disrupts the utilization of non-fermentable carbon sources in *C. albicans*. We hypothesized that defective carbon source utilization might also impair fungal adhesion. To test this, we replaced mannose (the carbon source in spider medium) with glucose, sucrose, ethanol, or acetate while maintaining the other experimental conditions. The crystal violet staining of *C. albicans* adhering to 12-well polystyrene plates revealed that the *qcr8Δ/Δ* mutant displayed reduced adhesion compared to the WT and *qcr8Δ/QCR8* strains across all tested carbon sources in spider medium (Fig. 4B and C). Mitochondria are important sites for carbon source processing, so we hypothesized that changes in mitochondrial function may be the cause of the above phenomena. Next, we explored the mitochondrial function of the *QCR8* mutant.

## The deletion of QCR8 leads to elevated ROS levels, reduces mitochondrial membrane potential, and decreases ATP and cAMP production in *C. albicans*

*QCR8*, an auxiliary subunit of complex III in the respiratory chain of *C. albicans*, is hypothesized to play a critical role in oxidative phosphorylation. Based on our previous findings showing that the *qcr8Δ/Δ* mutant exhibits impaired growth on non-fermentable carbon sources, we speculated that *QCR8* deletion disrupts redox homeostasis and mitochondrial ATP synthesis. To test this, spot assays were performed to assess the growth of the WT, *qcr8Δ/Δ*, and *qcr8Δ/QCR8* strains on YPD plates containing varying concentrations of $H_2O_2$. Compared to the WT and *qcr8Δ/QCR8* strains, the *qcr8Δ/Δ* mutant showed marked growth inhibition on plates supplemented with 0.03% $H_2O_2$ (Fig. 5A). Furthermore, intracellular ROS levels were significantly elevated in the *qcr8Δ/Δ* mutant (Fig. 5B and C), indicating the essential role of *QCR8* in maintaining redox balance. Previous studies have suggested that ROS accumulation in *C. albicans* compromises mitochondrial membrane potential ($\Delta\Psi m$), thereby reducing ATP production (25). As expected, we observed a decrease in the *qcr8Δ/Δ* mutant of $\Delta\Psi m$ (Fig. 5D). The generation of ATP is closely related to $\Delta\Psi m$ (26). Subsequent ATP quantification confirmed that *QCR8* deletion led to reduced ATP levels (Fig. 5E). As ATP hydrolysis generates cAMP, a key secondary messenger regulating signal transduction in *C. albicans*, we measured intracellular cAMP levels. The *qcr8Δ/Δ* mutant exhibited diminished cAMP

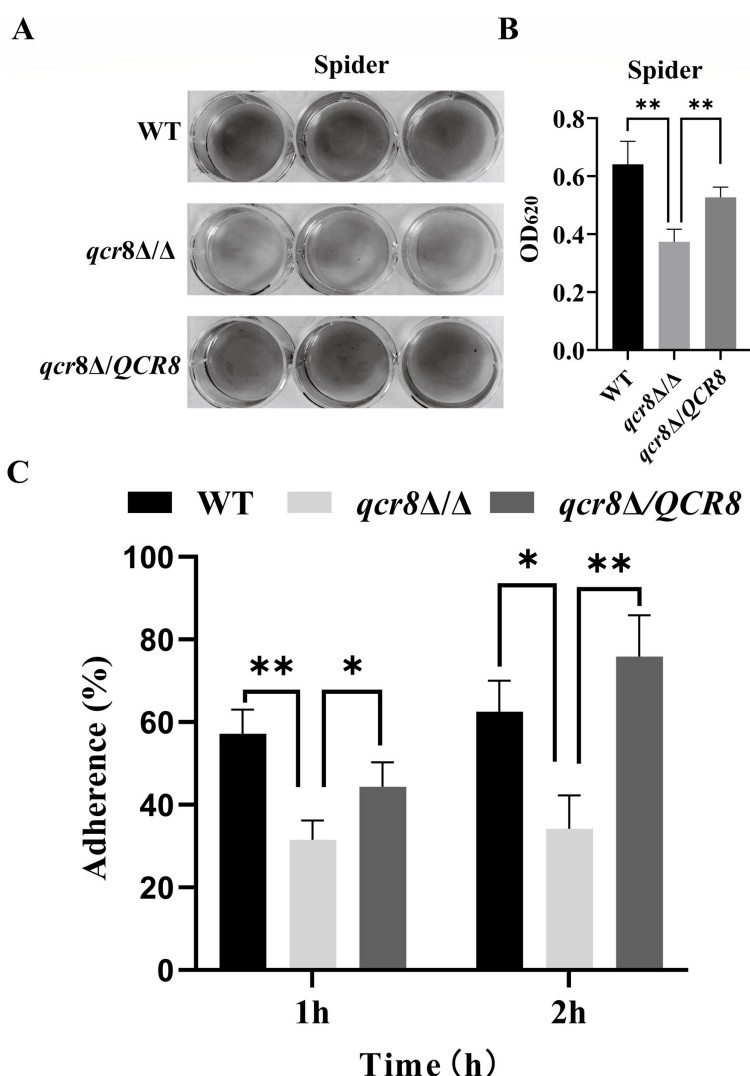

**FIG 3** *QCR8* affects the adhesion capacity of *C. albicans*. (A) *C. albicans* cell suspension was incubated in spider medium at 37°C for 90 min, followed by the crystal violet staining of fungal cells adhered to polystyrene surfaces in 12-well plates. (B) Absorbance at 620 nm was determined after destaining crystal violet-stained cells with acetic acid for 30 min. **$P < 0.01$. (C) The adherent-to-non-adherent ratios of fungal strains on HUVECs were measured after co-incubation with WT, *qcr8Δ/Δ*, and *qcr8Δ/QCR8* for 1 and 2 h, respectively. *$P < 0.05$, ** $P < 0.01$. Data are presented as mean ± SD.

production (Fig. 5F). To determine whether cAMP supplementation could rescue the adhesion defect caused by *QCR8* deletion, exogenous cAMP analogs (40 mM) were added to spider medium containing different carbon sources. The crystal violet staining of adhered cells in 12-well polystyrene plates demonstrated that exogenous cAMP partially restored the adhesion capacity of the *qcr8Δ/Δ* mutant (Fig. 5G and H). The above results indicate that the deletion of *QCR8* affects the mitochondrial function of *C. albicans*.

## The Ras/cAMP/PKA signaling pathway may contribute to the adhesion defect in the qcr8Δ/Δ mutant

The above results demonstrated reduced cAMP levels upon *QCR8* deletion. Given the central role of cAMP in the Ras/cAMP/PKA signaling pathway, we hypothesized that this pathway might mediate the adhesion impairment caused by *QCR8* deletion. To test this,

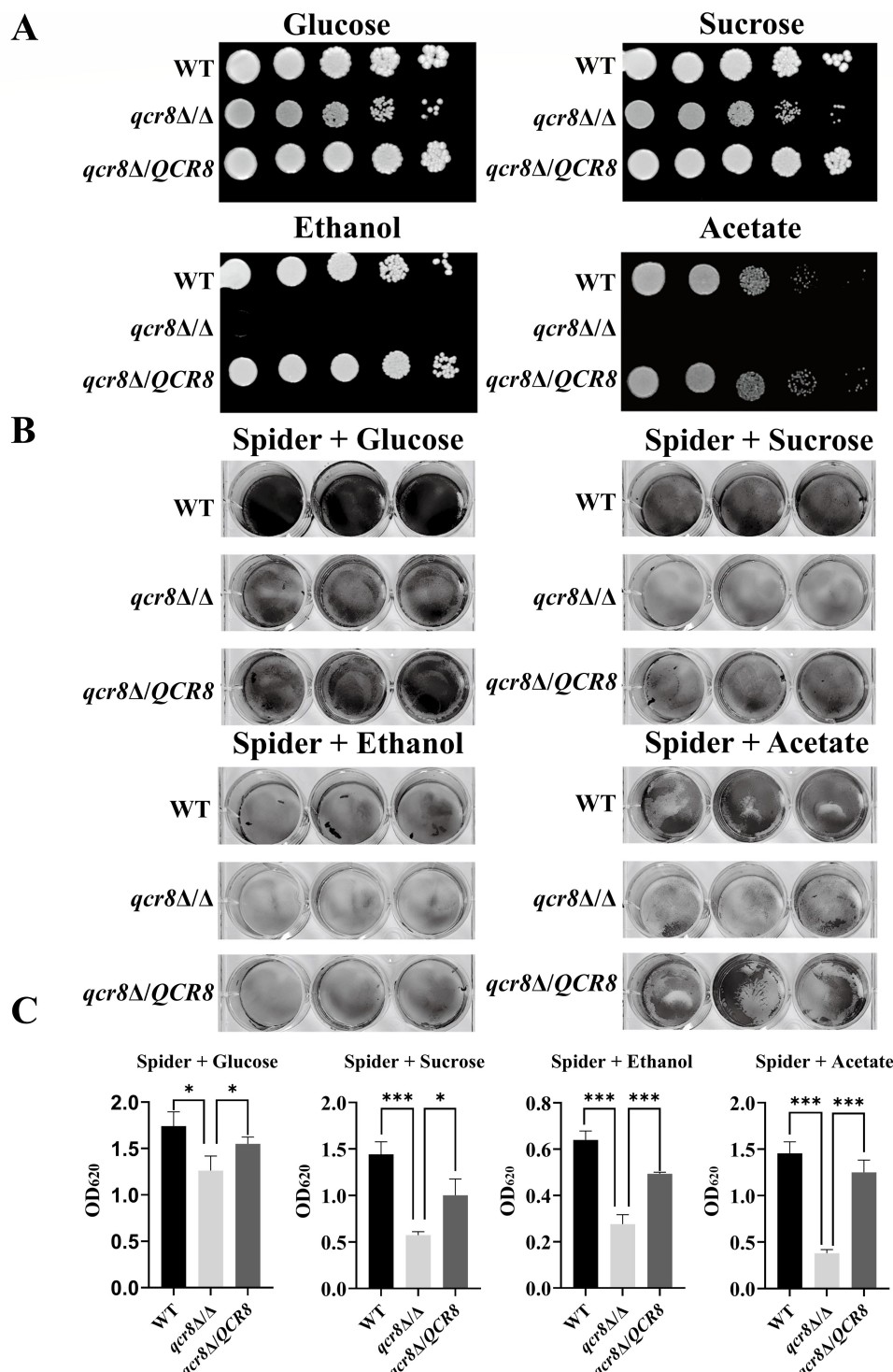

**FIG 4** *QCR8* affects carbon source utilization in *C. albicans*. (A) Overnight cultures of strains were washed with PBS, serially diluted, and spotted onto solid media containing glucose, sucrose, ethanol, or acetate as the sole carbon source. Plates were incubated at 30°C for 48 h, and growth was documented photographically. (B) In spider medium with mannitol replaced by glucose, sucrose, ethanol, or acetate, different strains adhered to polystyrene surfaces in 12-well plates were stained with crystal violet. (C) Absorbance at 620 nm was measured after destaining crystal violet-stained cells with acetic acid for 30 min. *P < 0.05, ***P < 0.001. Data are presented as mean ± SD.

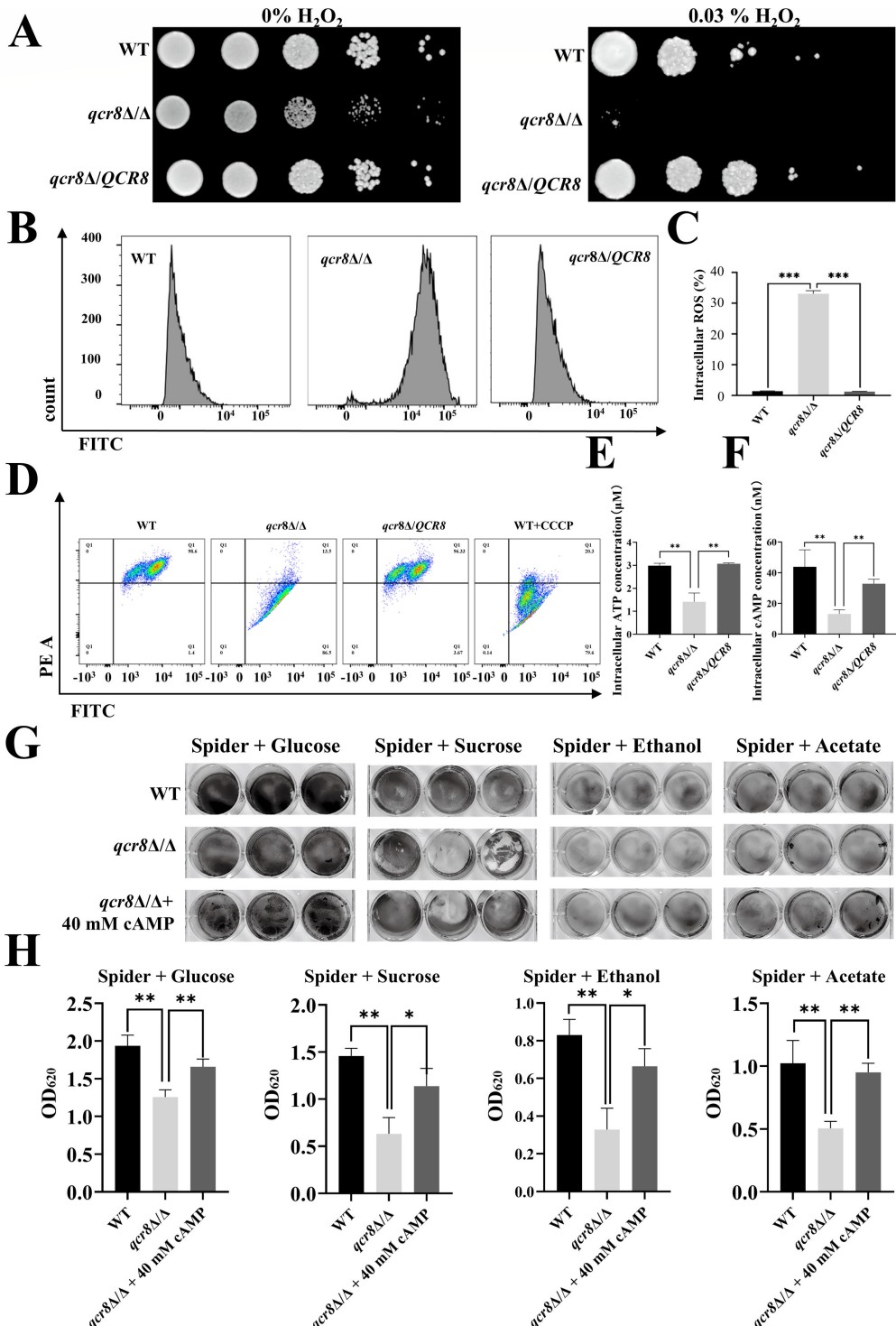

**FIG 5** The deletion of *QCR8* leads to elevated ROS levels, reduces mitochondrial membrane potential, and decreases ATP and cAMP production in *C. albicans*. (A) Overnight cultures of strains were washed with PBS, serially diluted, and spotted onto YPD plates containing varying concentrations of $H_2O_2$. The plates were incubated at 30°C for 48 h, and growth was documented photographically. (B) Intracellular ROS levels were quantified using an ROS detection assay kit. The fluorescence intensity (FITC channel) of different strains was measured via flow cytometry according to the instructions. (C) A statistical analysis of ROS levels from panel B. ***$P < 0.001$. (D) Mitochondrial membrane potential was assessed using a JC-1 assay kit. The fluorescence ratios (red/green) of JC-1 aggregates to monomers were analyzed via flow cytometry following the instructions. CCCP can induce a decrease in mitochondrial membrane potential in *Candida albicans*. (E) Intracellular ATP levels were determined using

Fig 5 (Continued)

an ATP test assay kit according to the instructions. **$P < 0.01$. (F) Intracellular cAMP levels were measured using a cAMP test assay kit following the instructions. (G) In spider medium with mannitol replaced by glucose, sucrose, ethanol, or acetate, adhered cells on polystyrene surfaces in 12-well plates were stained with crystal violet. An exogenous supplementation of 40 mM cAMP was included to assess the restoration of adhesion. (H) Absorbance at 620 nm was measured after destaining crystal violet-stained cells with acetic acid for 30 min.*$P < 0.05$, **$P < 0.01$. Data are presented as mean ± SD.

an RT-qPCR analysis was conducted, and it revealed that the expressions of key genes within the Ras/cAMP/PKA pathway were downregulated in the qcr8Δ/Δ mutant compared to those in the WT strain (Fig. 6A and B). These findings suggest that the adhesion defect associated with QCR8 deletion may be linked to the dysregulation of the Ras/cAMP/PKA signaling pathway.

## DISCUSSION

As a major pathogen of invasive fungal infections (27), elucidating the virulence mechanisms of C. albicans is crucial for antifungal drug development. This study reveals the critical role of QCR8, an accessory subunit of mitochondrial respiratory chain complex III, in regulating the virulence of C. albicans, establishing a regulatory network mediated by QCR8 that links energy metabolism, signal transduction, and adhesion capacity.

In this study, we found that Qcr8 and Qcr7 are adjacent in the spatial structure of complex III; however, knocking out QCR7 and QCR8 in C. albicans complex III results in different conformations. This indicates that Qcr7 and Qcr8 are two different functional accessory subunits of complex III in C. albicans. Therefore, the functions of Qcr8 deserve further investigation. Using Galleria mellonella and BALB/c mouse infection models, we confirmed that QCR8 deletion significantly attenuated the virulence of C. albicans. Notably, the complemented strain qcr8Δ/QCR8 restored virulence, confirming the specificity and irreplaceability of Qcr8 function. Histopathological and fungal burden analyses further showed that the QCR8 deletion strain exhibited a significantly reduced colonization capacity and an inflammation-inducing ability in hosts, consistent with the observed virulence phenotype. As the initial step of infection, adhesion directly influences fungal colonization and invasion (28). Through polystyrene surface adhesion assays and HUVEC adhesion models, we found that the QCR8 deletion strain displayed significant impairment in adhesion to both biotic and abiotic surfaces. Carbon source utilization assays revealed that the QCR8 deletion strain failed to grow on non-fermentative carbon sources. When the carbon source in spider medium was replaced with ethanol or acetate, the adhesion defect became more pronounced, suggesting that QCR8 may regulate adhesion by controlling the expression or conformation of adhesion molecules dependent on energy metabolism. Importantly, exogenous cAMP partially restored the adhesion capacity of the QCR8 deletion strain, indicating that cAMP, as a second messenger, mediates the association of these processes.

Mitochondria are considered the primary source of ROS in cells. The function and survival of eukaryotic cells depend on the mitochondrial H+ electrochemical gradient (Δp), which consists of the inner mitochondrial membrane potential (ΔΨmt) and pH gradient (ΔpH) (29). Studies have shown that ROS levels are closely linked to mitochondrial membrane potential, with the accumulation of ROS leading to its decrease (25). Meanwhile, research has indicated that a decline in membrane potential reduces mitochondrial ATP synthesis (26). As an accessory subunit of complex III, QCR8 deletion also resulted in abnormal ROS accumulation, decreased mitochondrial membrane potential, and reduced ATP production, consistent with previous findings, highlighting the core role of Qcr8 as an accessory subunit in maintaining the integrity of the electron transport chain and oxidative phosphorylation efficiency. Mechanistically, ROS accumulation may cause membrane potential collapse through interactions with electrons in the mitochondrial inner membrane, while oxidative stress-induced damage to cell membranes and nucleic acids may further impair the synthesis or secretion of

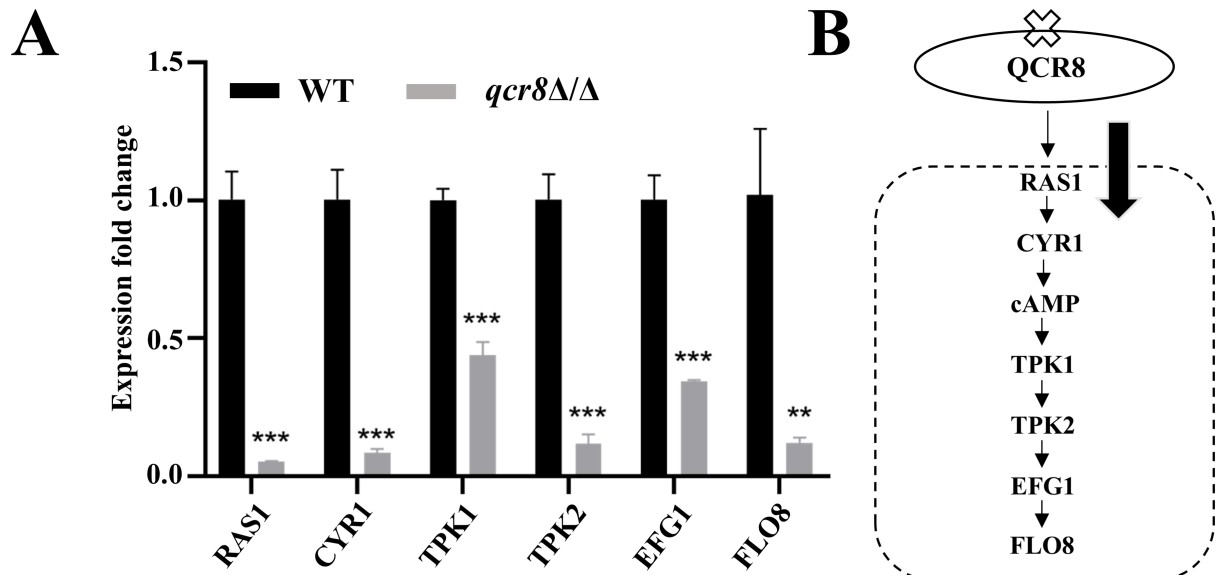

**FIG 6** The Ras/cAMP/PKA signaling pathway may contribute to the adhesion defect in the qcr8Δ/Δ mutant QCR8. (A) Total RNA was extracted from the WT and qcr8Δ/Δ strains. The expression levels of genes in the Ras/cAMP/PKA pathway were analyzed using RT-qPCR. **$P < 0.01$, ***$P < 0.001$. (B) The reduced adhesion capacity of the qcr8Δ/Δ strains may be mediated by the downregulation of the Ras/cAMP/PKA pathway. Data are presented as mean ± SD.

adhesion-related proteins, triggering a cascade of virulence attenuation. Notably, the *QCR8*-deficient strain exhibited more severe growth defects on non-fermentative carbon sources (such as ethanol and acetate), reflecting a specific impairment of mitochondrial respiratory function.

The Ras/cAMP/PKA signaling pathway is an important signaling pathway (30); it plays a central role in hyphal formation (13), biofilm development (31), and host adhesion (14). Adenylyl cyclase (Cyr1) in this pathway catalyzes the conversion of ATP into the ubiquitous second messenger cAMP (32). *QCR8* deletion impeded the interaction with Cyt1 (Fig. S9), reduced ATP levels, leading to decreased cAMP production, thereby downregulating the expression of the virulence-associated transcription factors *EFG1* and *FLO8* downstream of the Ras/cAMP/PKA signaling pathway. A reduced expression of *EFG1* and *FLO8* decreases the synthesis of adhesion proteins (such as the Als family and Sap proteases) (33, 34). Thus, the downregulation of genes in the Ras/cAMP/PKA signaling pathway directly explains the adhesion defect of the *QCR8*-deficient strain, forming a regulatory axis of "energy metabolism–signal transduction–virulence phenotype."

This study establishes a direct link between mitochondrial subunits and virulence regulation in *C. albicans*, demonstrating that *QCR8* controls adhesion and host infection by maintaining mitochondrial function and regulating the Ras/cAMP/PKA signaling pathway. These findings highlight the importance of non-structural mitochondrial subunits in pathogenicity. Given the growing challenge of antifungal resistance, targeting Qcr8 or its downstream pathways (such as developing QCR8-specific inhibitors or modulating cAMP levels to enhance host defense) represents a promising therapeutic strategy. However, several questions remain unanswered; for example, the mechanisms underlying the reduced expression of the upstream signaling molecules *Ras* and *CYR1* in the Ras/cAMP/PKA signaling cascade in the *QCR8* deletion strains are not fully explained, and the structural domains of Qcr8 in complex III need further validation through crystallography. Additionally, whether *QCR8* regulates virulence through non-mitochondrial pathways (such as direct interaction with signaling molecules) requires further exploration using proteomics and metabolomics approaches. Future research should focus on the molecular chaperone mechanisms of *QCR8* and its interactions with host immunity to identify precise targets for antifungal drug development.

In conclusion, this study identifies *QCR8* as a key hub linking energy metabolism, signal transduction, and adhesion capacity in the virulence regulation of *C. albicans*. These findings deepen our understanding of mitochondrial mechanisms in fungal pathogenesis and provide new insights for developing mitochondria-targeted antifungal therapies.

## ACKNOWLEDGMENTS

This work was supported by the National Nature Science Foundation of China (NSFC 32460049, 32060040, and 31760261), the Nanchang University First Affiliated Hospital Talent Cultivation Yang Fan Project (GCC-20250301), the Jiangxi Natural Science Foundation (20202BAB216045, 20202BAB206062, and 20204BCJL23054), the Double-Thousand Talent Program of Jiangxi Province (jxsq2023201019 and jxsq2023301110), the Training Plan for Academic and Technical Leaders of Major Disciplines in Jiangxi Province-Youth Talent Project (20212BCJ23036), and the Jiangxi Province Graduate Student Innovation Fund Project (YC2024-S045).

## AUTHOR AFFILIATIONS

[1]Department of Laboratory and Gaoxin Branch of The First Affiliated Hospital, Jiangxi Medical College, Nanchang University, Nanchang, China
[2]Nanchang Key Laboratory of Diagnosis of Infectious Diseases, Gaoxin Branch of The First Affiliated Hospital, Jiangxi Medical College, Nanchang University, Nanchang, China
[3]Medical Experimental Teaching Center, School of Basic Medical Sciences, Jiangxi Medical College, Nanchang University, Nanchang, China
[4]Key Laboratory of Prevention and Treatment of Cardiovascular and Cerebrovascular Diseases, Ministry of Education, School of Basic Medicine, Gannan Medical University, Ganzhou, Jiangxi, China

## AUTHOR ORCIDs

Qianjun Zhao http://orcid.org/0009-0004-2839-6227
Xiaotian Huang http://orcid.org/0000-0003-1145-8386
Qiong Liu http://orcid.org/0000-0002-8950-6133
Lingbing Zeng http://orcid.org/0000-0001-9404-6605

## DATA AVAILABILITY

All data are available from authors upon reasonable requirement.

## ETHICS APPROVAL

The ethics committee of Nanchang University approved this study (approval number CDYFY-IACUC-202407QR164).

## ADDITIONAL FILES

The following material is available online.

### Supplemental Material

**Figures S1 to S10; Tables S1 and S2 (Spectrum01672-25-s0001.docx).** Additional experiments.

### Open Peer Review

**PEER REVIEW HISTORY (review-history.pdf).** An accounting of the reviewer comments and feedback.

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
