## [Reviewer comments · Microbiology Spectrum]

Microbiology Spectrum

Mitochondrial complex III subunit Qcr8 regulates the virulence and adhesion of *Candida albicans* by modulating mitochondrial function

Lingbing Zeng, Qianjun Zhao, Xiaotian Huang, Qiong Liu, Yanli Cao, Xiaomin Yu, Yanling Liu, Niya Hu, Zhenning Han, Junjun Tan, Xuan Li, Yemin zhang, Kai Wang, and Yuting Li

Corresponding Author(s): Lingbing Zeng, The First Affiliated Hospital of Nanchang University

Review Timeline:

Submission Date:	May 30, 2025
Editorial Decision:	July 19, 2025
Revision Received:	September 28, 2025
Editorial Decision:	October 12, 2025
Revision Received:	November 2, 2025
Accepted:	November 20, 2025

Editor: Jonathan Snow

Reviewer(s): The reviewers have opted to remain anonymous.

Transaction Report:

DOI: <https://doi.org/10.1128/spectrum.01672-25>

Re: Spectrum01672-25 (**Mitochondrial complex III subunit Qcr8 regulates the virulence and adhesion of *Candida albicans* by modulating mitochondrial function**)

Dear Dr. Lingbing Zeng:

Thank you for the privilege of reviewing your work. Below you will find my comments, instructions from the Spectrum editorial office, and the reviewer comments. Although mostly positive about your manuscript, both reviewers have concerns that preclude publication of this manuscript in its current form. In particular, both reviewers would like clarification on specific methodological details. In addition, both reviewers suggested specific experiments that they felt would strengthen support for the conclusions drawn. Please make sure to address all of the reviewer concerns including these major points in any resubmission.

Revision Guidelines

Sincerely,
Jonathan Snow
Editor
Microbiology Spectrum

Reviewer #1 (Comments for the Author):

Zhao et.al. 2025 investigates Qcr8 as an accessory subunit of the mitochondrial complex III for *C. albicans* full virulence.

The study elucidates that Qcr8 is an accessory subunit of the mitochondrial complex III; the subunit plays a role in regulation of

virulence; Qcr8 promotes virulence via the Ras/cAMP/PKA pathway and Qcr8 is a potential therapeutic target for treating *C. albicans* interactions. In the study the authors' focus on QCR8 effects on adhesion capacity, carbon source utilization and expression of genes associated with Ras/PKA/cAMP pathway. Their intended goal is to uncover the role of QCR8 in *C. albicans* pathogenesis, thus providing possible targets and theoretical foundations for new antifungal drug development. The manuscript is interesting and provides insights into the roles of QCR8 (as distinguished from QCR7) in *C. albicans* virulence. I have some suggestions for the authors' consideration:

Comments on the Results:

Figure 1:

Line 348 to 354 has some inconsistencies and should be rewritten. It would be helpful to know what the score range for the iPTM scores and pLDDT values are, to understand relevance of the scores reported in the study for the deletion strains. In addition, in line 342, it was originally reported that the complete complex has 31 hydrogen bonds, but it is later stated in line 348, "In terms of the number of hydrogen bonds the complete complex had 4...." This needs to be explained further as it creates confusion.

As the authors later mentioned in the Discussion, I do suggest that co-immunoprecipitation experiments may add to the studies showing the interaction partners of Qcr7, Qcr8, Cyt1, Cytb in the various strain backgrounds.

Figure 2A/2B: Distinction markers on the survival curves, showing WT, deletion and complemented strains, should be made clearer to interpret the graph.

Figure 3B/C: Bar graph shading between *qcr8Δ/Δ* and *qcr8Δ/QCR8* is too similar and should be adjusted for better interpretation.

Figure 3C: No speculative explanation was given for the % in adherence between the WT and the *qcr8Δ/QCR8* strains.

Figure 4C: Bar graph shading between *qcr8Δ/Δ* and *qcr8Δ/QCR8* is too similar and should be adjusted for better interpretation.

Figure 5C/F/H: Bar graph shading between *qcr8Δ/Δ* and *qcr8Δ/QCR8* is too similar and should be adjusted for better interpretation.

Comments on the Discussion:

It might be important to consider that the effects seen in the *qcr8Δ/Δ* mutants maybe indirect through an interacting protein. The authors should evaluate this possibility by repeating some of the experiments in this study with various double mutants of the proteins in the complex.

Reviewer #2 (Comments for the Author):

In this manuscript, Zhao and colleagues examine the function of the Qcr8 mitochondrial accessory protein in growth, stress tolerance, adhesion, and virulence. They use protein structure predictions to develop a hypothesis that Qcr7 and Qcr8 are not redundant proteins, and then generate the *qcr8Δ/Δ* mutant strain in *C. albicans* to test the function of this protein. The mutant strain is less able to cause disease in two animal models of infection, and is also less able to grow and adhere to surfaces.

The main issue is that the mutant is also sicker under rich media conditions with plenty of glucose. This defect in growth may explain the adhesion, stress tolerance, and virulence defects.

Minor points:

Figure 1, please orient all models in the same direction and indicate the subunits.

Figure 3, please show the wells prior to washing - if the mutant grew more poorly, it is not a specific adhesion defect. The relative OD might be a better measure. Figure 3C legend - please correct bacterial to fungal

Figure 4: again, if the mutant cannot grow on the different carbon sources, the adhesion defect might be explained purely by a lack of growth. The spot plates in A are nicely done, but the adhesion in B and C might be confounded.

Figure 5D, please include a control of WT with a mitochondrial poison to demonstrate specificity of the JC-1 signal.

Zhao et.al. 2025 investigates Qcr8 as an accessory subunit of the mitochondrial complex III for *C. albicans* full virulence.

The study elucidates that Qcr8 is an accessory subunit of the mitochondrial complex III; the subunit plays a role in regulation of virulence; Qcr8 promotes virulence via the Ras/cAMP/PKA pathway and Qcr8 is a potential therapeutic target for treating *C. albicans* interactions. In the study the authors' focus on *QCR8* effects on adhesion capacity, carbon source utilization and expression of genes associated with Ras/PKA/cAMP pathway. Their intended goal is to uncover the role of *QCR8* in *C. albicans* pathogenesis, thus providing possible targets and theoretical foundations for new antifungal drug development.

The manuscript is interesting and provides insights into the roles of *QCR8* (as distinguished from *QCR7*) in *C. albicans* virulence. I have some suggestions for the authors' consideration:

Comments on the Results:

Figure 1:

Line 348 to 354 has some inconsistencies and should be rewritten. It would be helpful to know what the score range for the iPTM scores and pLDDT values are, to understand relevance of the scores reported in the study for the deletion strains.

In addition, in **line 342**, it was originally reported that the complete complex has **31 hydrogen bonds**, but it is later stated in **line 348**, "In terms of the number of hydrogen bonds the complete complex had 4...." This needs to be explained further as it creates confusion.

As the authors later mentioned in the Discussion, I do suggest that co-immunoprecipitation experiments may add to the studies showing the interaction partners of Qcr7, Qcr8, Cyt1, Cytb in the various strain backgrounds.

Figure 2A/2B: Distinction markers on the survival curves, showing WT, deletion and complemented strains, should be made clearer to interpret the graph.

Figure 3B/C: Bar graph shading between *qcr8Δ/Δ* and *qcr8Δ/QCR8* is too similar and should be adjusted for better interpretation.

Figure 3C: No speculative explanation was given for the % in adherence between the WT and the *qcr8Δ/QCR8* strains.

Figure 4C: Bar graph shading between *qcr8Δ/Δ* and *qcr8Δ/QCR8* is too similar and should be adjusted for better interpretation.

Figure 5C/F/H: Bar graph shading between *qcr8Δ/Δ* and *qcr8Δ/QCR8* is too similar and should be adjusted for better interpretation.

Comments on the Discussion:

It might be important to consider that the effects seen in the *qcr8Δ/Δ* mutants maybe indirect through an interacting protein. The authors should evaluate this possibility by repeating some of the experiments in this study with various double mutants of the proteins in the complex.

Dear editors,

We feel great thanks for your professional review on our manuscript entitled

“Mitochondrial complex III subunit Qcr8 regulates the virulence and adhesion of *Candida albicans* by modulating mitochondrial function” (Spectrum01672-25).

As you are concerned, there are several problems that need to be addressed. We have made corrections to our draft and supplemented extra data to make our results convincing according to your useful comments. The reviewers' comments are laid out in italicized font and specific concerning questions have been numbered. Our response is given normal **BLUE** font and changes/ additions to the **Marked-up Manuscript and Supplement material** are shown in the **RED** text. Detailed corrections are listed below.

Reviewer #1:

The study elucidates that Qcr8 is an accessory subunit of the mitochondrial complex III; the subunit plays a role in regulation of virulence; Qcr8 promotes virulence via the Ras/cAMP/PKA pathway and Qcr8 is a potential therapeutic target for treating C. albicans interactions. In the study the authors' focus on QCR8 effects on adhesion capacity, carbon source utilization and expression of genes associated with Ras/PKA/cAMP pathway. Their intended goal is to uncover the role of QCR8 in C. albicans pathogenesis, thus providing possible targets and theoretical foundations for new antifungal drug development.

The manuscript is interesting and provides insights into the roles of QCR8 (as distinguished from QCR7) in C. albicans virulence. I have some suggestions for the authors' consideration:

Comments on the Results:

Comment 1:

Line 348 to 354 has some inconsistencies and should be rewritten. It would be helpful to know what the score range for the iPTM scores and pLDDT values are, to understand relevance of the scores reported in the study for the deletion strains.

Response: Thank you for your comment. We have added supplementary explanations regarding iPTM, pTM, and pLDDT in the revised manuscript (lines 332-342). Briefly, the iPTM score is used to assess the accuracy of multi-polymer interface predictions, ranging from 0 to 1, with higher values indicating greater accuracy. This score is primarily applied to predict complex structures. The pTM score reflects the global confidence of the overall structural prediction, also ranging from 0 to 1, with higher values indicating greater reliability. Generally, scores above 0.8 suggest a high level of confidence. The pLDDT score represents the residue-level prediction accuracy, ranging from 0 to 100. Scores below 50 indicate very low accuracy or an intrinsically disordered protein; scores below 70 indicate low confidence; and scores above 90 indicate extremely high confidence.

Comment 2:

In line 342, it was originally reported that the complete complex has 31 hydrogen bonds, but it is later stated in line 348, "In terms of the number of hydrogen bonds the complete complex had 4..." This needs to be explained further as it creates confusion.

Response: Thank you for pointing out this error, and we sincerely apologize for the oversight. It meant the number of salt bridges, not hydrogen bonds. We have corrected this in the revised manuscript, changing "hydrogen bonds" to "salt bridges" (line 362-365).

Comment 3:

As the authors later mentioned in the Discussion, I do suggest that co-immunoprecipitation experiments may add to the studies showing the interaction partners of Qcr7, Qcr8, Cyt1, Cytb in the various strain backgrounds.

Response: Thank you for your insightful suggestion. To investigate whether Qcr8 interacts with Qcr7, Cyt1, and Cytb, we constructed the strains SN152-Qcr8-3HA, SN152-Qcr7-3Flag, SN152-Cyt1-3Flag, Qcr8-3HA-qcr7-3Flag, and Qcr8-3HA-Cyt1-3Flag. Despite more than three repeated attempts, we were unable to

amplify *cytb* by PCR and therefore could not construct SN152-Cytb-3Flag and Qcr8-3HA-Cytb-3Flag. Western blot analysis confirmed the successful construction of SN152-Qcr8-3HA, SN152-Qcr7-3Flag, SN152-Cyt1-3Flag, Qcr8-3HA-qcr7-3Flag, and Qcr8-3HA-Cyt1-3Flag (Figure S8). Subsequent co-immunoprecipitation assays demonstrated that Qcr7 and Cyt1 interact with Qcr8, which we hypothesize reflects the fact that these three proteins are all subunits of *Candida albicans* complex III. (Figure S9)

Figure S8

Figure S9

Comment 4:

Figure 2A/2B: Distinction markers on the survival curves, showing WT, deletion and complemented strains, should be made clearer to interpret the graph.

Response: Thank you for your valuable suggestion. We fully agree with your concern regarding the readability of strain markers (WT, deletion mutant, and complemented strain) on the survival curves. Based on your comment, we found that the markers for the deletion mutant and the PBS group were prone to confusion. To address this issue, we have redesigned the marker scheme for these groups to ensure a clearer visual distinction for readers.

Original: (Figure 2A)

Revised: (Figure 2A):

Original: (Figure 2B):

Revised: (Figure 2B):

Comment 5:

Figure 3B/C: Bar graph shading between *qcr8Δ/Δ* and *qcr8Δ/QCR8* is too similar and should be adjusted for better interpretation.

Response: Thank you for your insightful suggestion. We have adjusted the color schemes for the *qcr8Δ/Δ* and *qcr8Δ/QCR8* strains to improve the visual distinction between groups. The original and revised figures are provided below for your review.

Original: (Figure 3B)

Revised: (Figure 3B)

Original: (Figure 3C)

Revised: (Figure 3C):

Comment 6:

Figure 3C: No speculative explanation was given for the % in adherence between the WT and the *qcr8Δ/QCR8* strains.

Response: Thank you for your comment. In the Materials and Methods section (lines 245–246), we specifically defined the meaning of “% in adherence between the WT and the *qcr8Δ/QCR8* strains.” Briefly, this “%” represents the proportion of adherent cells, calculated as follows: After washing off the non-adherent cells with PBS, the remaining adherent cells were treated with 0.5% Triton X-100 lysis buffer, diluted, plated, and incubated for 48 hours to determine the colony count (treatment group). Without PBS washing to remove non-adherent cells, the cells were directly diluted, plated, and incubated for 48 hours to determine the colony count (control group). The adherence rate (%) was then calculated using the following formula:

$$\text{Adherence rate (\%)} = (\text{colony count of treatment group} / \text{colony count of control group}) \times 100\%.$$

Comment 7:

Figure 4C: Bar graph shading between *qcr8Δ/Δ* and *qcr8Δ/QCR8* is too similar and should be adjusted for better interpretation.

Response: Thank you for your insightful suggestion. We have updated the color schemes for the *qcr8Δ/Δ* and *qcr8Δ/QCR8* strains to improve the visual distinction between groups. The original and revised figures are provided below for your review.

Original: (Figure 4C)

Revised: (Figure 4C):

Comment 8:

Figure 5C/F/H: Bar graph shading between *qcr8Δ/Δ* and *qcr8Δ/QCR8* is too similar and should be adjusted for better interpretation.

Response: Thank you for your insightful suggestion. We have updated the color

schemes for the $qcr8\Delta/\Delta$ and $qcr8\Delta/QCR8$ strains to improve the visual distinction between groups. The original and revised figures are provided below for your review.

Original: (Figure 5C)

Revised: (Figure 5C):

Original: (Figure 5E)

Revised: (Figure 5E):

Original: (Figure 5F)

Revised: (Figure 5F):

Original: (Figure 5H)

Revised: (Figure 5H):

Comments on the Discussion:

Comment 1:

It might be important to consider that the effects seen in the $qcr8\Delta/\Delta$ mutants may be indirect through an interacting protein. The authors should evaluate this possibility by repeating some of the experiments in this study with various double mutants of the proteins in the complex.

Response: Thank you for your insightful suggestion. To determine whether Qcr7 and Cyt1 affect the function of Qcr8, we overexpressed *qcr7* and *cyt1* in the $qcr8\Delta/\Delta$ strain to achieve the same objective. RT-qPCR analysis confirmed that the expression levels of *qcr7* and *cyt1* were significantly higher in $qcr8\Delta/\Delta$ -*qcr7*^{OE} and $qcr8\Delta/\Delta$ -*cyt1*^{OE} compared with $qcr8\Delta/\Delta$, indicating successful construction of these strains (Figure S10A).

We then conducted similar experiments. Importantly, ATP and cAMP levels in $qcr8\Delta/\Delta$ -*qcr7*^{OE} were not restored, whereas ATP and cAMP levels in $qcr8\Delta/\Delta$ -*cyt1*^{OE} differed significantly compared with $qcr8\Delta/\Delta$. Furthermore, in the adhesion assay, the adhesion capacity of $qcr8\Delta/\Delta$ -*qcr7*^{OE} showed no evident recovery, whereas the adhesion capacity of $qcr8\Delta/\Delta$ -*cyt1*^{OE} was markedly restored relative to $qcr8\Delta/\Delta$.

Taken together, these findings demonstrate that Cyt1, as a catalytic subunit of *Candida albicans* complex III, interacts with the accessory subunit Qcr8 and influences its function. In contrast, Qcr7, as an accessory subunit, does not interact with Qcr8 and does not affect its function.

Figure S10

Reviewer #2:

*In this manuscript, Zhao and colleagues examine the function of the Qcr8 mitochondrial accessory protein in growth, stress tolerance, adhesion, and virulence. They use protein structure predictions to develop a hypothesis that Qcr7 and Qcr8 are not redundant proteins, and then generate the qcr8 Δ/Δ mutant strain in *C. albicans* to test the function of this protein. The mutant strain is less able to cause disease in two animal models of infection, and is also less able to grow and adhere to surfaces.*

The main issue is that the mutant is also sicker under rich media conditions with plenty of glucose. This defect in growth may explain the adhesion, stress tolerance, and virulence defects.

Comment 1:

Figure 1, please orient all models in the same direction and indicate the subunits.

Response: Thank you for your valuable suggestion. We have aligned all models in the same orientation and labeled each subunit accordingly. The original and revised versions are shown in the figure below.

Original: (Figure 1 C-F)

Revised: (Figure 1 C-F):

Comment 2:

Figure 3, please show the wells prior to washing - if the mutant grew more poorly, it is not a specific adhesion defect. The relative OD might be a better measure. Figure 3C legend - please correct bacterial to fungal

Response: Thank you for your valuable suggestion. In accordance with your comments, we performed additional experiments in which different carbon sources were incorporated into the culture medium and incubated at 37 °C for 90 minutes (these conditions correspond to our adhesion assay as detailed in the Materials and Methods section). We then analyzed samples in which the culture medium was removed without PBS washing (Figure S6). Under these unwashed conditions, no significant differences were observed between *qcr8Δ/Δ* and WT across the various media. However, after washing to remove unattached cells, significant differences emerged between *qcr8Δ/Δ* and WT (Figure 4B).

Furthermore, to compare the growth rates of *qcr8Δ/Δ* and WT in the presence of different carbon sources during the 90-minute incubation at 37 °C, we generated growth curves (Figure S7). These showed no significant differences in growth rates between *qcr8Δ/Δ* and WT during this period. Therefore, we can rule out the possibility that growth-rate differences account for the observed differences in adhesion.

changed from “bacterial” to “fungal” (Line 717)

Figure S6

Figure 3A

Figure S7

Spider

Spider + Glucose

Spider + Sucrose

Spider + Ethanol

Spider + Acetate

Comment 3:

Figure 4: again, if the mutant cannot grow on the different carbon sources, the adhesion defect might be explained purely by a lack of growth. The spot plates in A are nicely done, but the adhesion in B and C might be confounded.

Response: Thank you for your valuable suggestion. As stated in the Materials and Methods section, the incubation time for the plate assay is 48 hours, during which we observed that the mutant strain exhibited growth defects. In contrast, the incubation time for the adhesion assay is only 90 minutes. To determine whether there were any differences in the growth rates of the mutant and wild-type strains during this period, we measured their growth rates under the same conditions. As shown in the growth curve (Figure S7), no significant differences were observed between the mutant and wild-type strains during the adhesion period, thereby ruling out the influence of growth defects.

Figure S7

Comment 4:

Figure 5D, please include a control of WT with a mitochondrial poison to demonstrate specificity of the JC-1 signal.

Response: Thank you for your valuable suggestion. In accordance with your recommendation, we added the inhibitor CCCP (carbonyl cyanide m-chlorophenyl hydrazone) to the WT strain. This compound induces a decrease in mitochondrial membrane potential in *Candida albicans*, thereby confirming the specificity of the JC-1 signal. The original and revised figures are shown below.

Original: (Figure 5D)

Revised: (Figure 5 D):

Re: Spectrum01672-25R1 (**Mitochondrial complex III subunit Qcr8 regulates the virulence and adhesion of *Candida albicans* by modulating mitochondrial function**)

Dear Dr. Lingbing Zeng:

Thank you for the privilege of reviewing your work. Below you will find my comments, instructions from the Spectrum editorial office, and the reviewer comments. As you will see, one reviewer is fully satisfied with the efforts made by your team, while the other reviewer requires an additional control to make the manuscript ready for publication.

Revision Guidelines

Sincerely,
Jonathan Snow
Editor
Microbiology Spectrum

Reviewer #2 (Comments for the Author):

For the new figure S9, the HA pulldown also needs to include a strain that only has the FLAG-Qcr7 to show that the FLAG-tagged protein is not interacting with the HA antibody nonspecifically. In addition, providing the full blots as opposed to cropped bands in the response to reviewers would be helpful in judging the outcome of this experiment.

Dear editors,

We feel great thanks for your professional review on our manuscript entitled **“Mitochondrial complex III subunit Qcr8 regulates the virulence and adhesion of *Candida albicans* by modulating mitochondrial function” (Spectrum01672-25R2).**

As you are concerned, there are several problems that need to be addressed. We have made corrections to our supplemented extra data to make our results convincing according to your useful comments. We have made corrections to our draft and supplemented extra data to make our results convincing according to your useful comments. The reviewers' comments are laid out in italicized font and specific concerning questions have been numbered. Our response is given normal **BLUE** font and changes / additions to the **Supplement material** are shown in the **RED** text.

Detailed corrections are listed below.

Reviewer #2

For the new figure S9, the HA pulldown also needs to include a strain that only has the FLAG-Qcr7 to show that the FLAG-tagged protein is not interacting with the HA antibody nonspecifically. In addition, providing the full blots as opposed to cropped bands in the response to reviewers would be helpful in judging the outcome of this experiment.

Comment 1:

For the new figure S9, the HA pulldown also needs to include a strain that only has the FLAG-Qcr7 to show that the FLAG-tagged protein is not interacting with the HA antibody nonspecifically.

Response: Thank you for your insightful suggestion. According to your suggestion, we included an additional control group. The results showed that the FLAG-tagged protein did not exhibit nonspecific binding to the HA antibody, thereby supporting our

previous conclusion that Qcr8 interacts with Qcr7 and Cyt1.

Figure S8

Figure S9

Comment 2:

In addition, providing the full blots as opposed to cropped bands in the response to reviewers would be helpful in judging the outcome of this experiment.

Response: Thank you for your insightful suggestion. According to your suggestion, we repeated the corresponding Western blot experiment. Using the protein elution method, proteins from the same group were exposed on the same NC membrane. We have provided the full blots along with the protein markers to help you better assess our experimental results

Re: Spectrum01672-25R2 (**Mitochondrial complex III subunit Qcr8 regulates the virulence and adhesion of *Candida albicans* by modulating mitochondrial function**)

Dear Dr. Lingbing Zeng:

Your manuscript has been accepted, and I am forwarding it to the ASM production staff for publication. Your paper will first be checked to make sure all elements meet the technical requirements. ASM staff will contact you if anything needs to be revised before copyediting and production can begin. Otherwise, you will be notified when your proofs are ready to be viewed.

Sincerely,
Jonathan Snow
Editor
Microbiology Spectrum